# Twin-field quantum key distribution without optical frequency dissemination

Lai Zhou [1], Jinping Lin [1], Yumang Jing [1] & Zhiliang Yuan [1]✉

Twin-field (TF) quantum key distribution (QKD) has rapidly risen as the most viable solution to long-distance secure fibre communication thanks to its fundamentally repeater-like rate-loss scaling. However, its implementation complexity, if not successfully addressed, could impede or even prevent its advance into real-world. To satisfy its requirement for twin-field coherence, all present setups adopted essentially a gigantic, resource-inefficient interferometer structure that lacks scalability that mature QKD systems provide with simplex quantum links. Here we introduce a technique that can stabilise an open channel without using a closed interferometer and has general applicability to phase-sensitive quantum communications. Using locally generated frequency combs to establish mutual coherence, we develop a simple and versatile TF-QKD setup that does not need service fibre and can operate over links of 100 km asymmetry. We confirm the setup's repeater-like behaviour and obtain a finite-size rate of 0.32 bit/s at a distance of 615.6 km.

Communication at the single photon level enables quantum key distribution (QKD) to achieve a revolutionary milestone in information security, allowing two distant users to establish a cryptographic key with verifiable secrecy[1,2]. Decades' development has advanced fibre-based QKD systems to a maturity level that is suitable for showcasing long-term, uninterrupted services in real-world networks[3–6]. In such systems, quantum signals experience the loss of the entire link and thus their maximally achievable rates scale linearly with the channel transmittance ($\eta$)[7–9]. This rate-loss scaling leads to a prohibitively low rate for long haul links, while such links often bear strategic importance for connecting metropolitan cities. Theoretically, quantum repeaters[10,11] can improve this scaling to $\eta^{\frac{1}{N+1}}$ with $N$ intermediate nodes and thus enable secure communications over arbitrarily long distances, but require technologies that are yet to become practical.

Twin-field (TF) QKD protocol[12] was recently proposed for practical long-haul quantum communications. Similarly to the form of QKD that achieves measurement-device-independence[13,14] (MDI), it uses an intermediate measurement node that halves the signal transmission loss, but extracts the information from the first-order interference rather than two-photon coincidence so as to gain the advantageous repeater-like rate-loss scaling of $\sqrt{\eta}$. Security against general attacks was proven for protocol variants[15–21], among which sending-not-sending (SNS)[17] and no-phase-post selection (NPP)[18–20] remove

partially the need for phase slice reconciliation and thus improve key generation efficiency. An exciting collection of experiments[22–33] has successfully validated TF-QKD's superior rate-loss scaling and repeatedly broken the communication distance record, which now stands at 833 km[32].

With above achievements, the primary experimental focus should now turn to addressing TF-QKD's implementation complexity, which would otherwise impair its practical deployment. Due to the stringent requirement for twin-field phase tracking, all existing long-haul TF-QKD setups have to adopt essentially a gigantic Mach–Zehnder inteferometer (MZI) configuration with half of its fibre used for optical frequency dissemination (Fig. 1a). Use of service fibre brings two severe drawbacks. First, the setups are fibre-resource inefficient and require additional frequency locking hardware and often optical amplifiers along the service fibre. Second, the closed fibre configuration is inherently incompatible with optical switching, which could restrict TF-QKD's scalability into a larger network like other QKD systems[34,35]. Using Sagnac interferometer permits a simple TF-QKD ring network[36], but its long-haul capability could be obstructed by noise contamination due to counter-propagating signals of strong intensity disparity. Ideally, the twin-field phase could be stabilised without using a closed interferometer so as to reach a simple setup

[1]Beijing Academy of Quantum Information Sciences, Beijing 100193, China. ✉e-mail: yuanzl@baqis.ac.cn

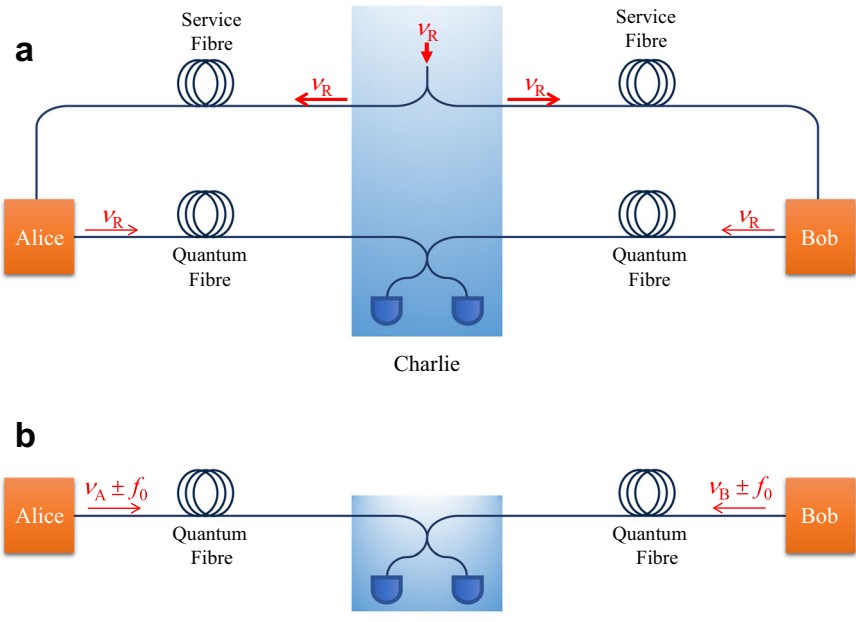

**Fig. 1 | Schematics of TF-QKD setups.** In TF-QKD, the users (Alice and Bob) communicate with each other by sending encoded quantum signals at the single-photon level to the intermediate node (Charlie), who measures the interference using two single photon detectors. The protocol's stringent requirement for phase stability has rendered all existing setups to adopt a closed interferometer configuration, which is resource-inefficient and inflexible. **a** Existing Mach–Zehnder interferometer setup[28,31,32]. Alice and Bob inherit a common optical frequency $\nu_R$ that is disseminated by Charlie via the long service fibres. **b** Open channel setup. Alice and Bob locally generate their own optical frequencies and their coherent side-bands of $\nu_A \pm f_0$ and $\nu_B \pm f_0$, with a nominally identical microwave frequency offset $f_0$. One side-band is used to reconcile the laser frequency difference ($\Delta \nu = \nu_A - \nu_B$), while the other is allocated quantum signal encoding. The open scheme eliminates the need for the service fibre and the optical frequency locking hardware, and supports asymmetric links.

(Fig. 1b) sharing an identical fibre configuration as MDI-QKD. We note that new MDI-QKD variants were recently proposed to allow repeater-like rate-loss scaling via post-detection time-bin pairing[37,38].

To perform TF-QKD, it is necessary to ensure stable interference at the intermediate node (Charlie) between signals transmitted by two remote users (Alice and Bob). The phase difference ($\phi$) between their signals evolves as

$$\frac{d\phi}{dt} = 2\pi \left( \Delta\nu + \frac{\nu}{c} \frac{d\Delta(nL)}{dt} \right), \tag{1}$$

where $\Delta\nu$ is the difference between the users' laser frequencies ($\nu$), $c$ light speed in vacuum, $n$ is the refractive index of the fibre, and $\Delta(nL)$ the optical length difference between the users' fibres to Charlie. The fibre term $\frac{\nu}{c}\frac{d\Delta(nL)}{dt}$ contributes typically a few kHz to the phase instability for a fibre link of several hundred kilometres[12,28]. This kHz drift alone can be corrected for with a feedback loop of a sub-MHz bandwidth using a low-level reference signal, without scattering overwhelming noise photons and thus crucially maintaining the quantum channel intact. However, the laser term is more problematic. Free-running lasers have unsatisfying long-term stability, with daily frequency drift often in the region of 10–100 MHz, although some can offer instantaneous linewidth of 1 kHz. Referencing to a high-fineness cavity helps, but the cavity itself drifts. Consequently, TF-QKD setups to date resort to sending strong laser signals to synchronise the users' lasers to enforce $\Delta\nu = 0$ and hence require a separate service fibre channel to avoid contamination to the quantum link (see Fig. 1a). The resulting setup has a closed fibre configuration.

In this work, we develop a scheme to stabilise an open quantum link between two distant users that share no prior mutual coherence and are separated by hundreds kilometres of single mode fibre. Using this scheme we have achieved a drastically simplified

and versatile TF-QKD setup, capable of supporting asymmetric links, that needs neither dissemination of optical frequencies nor its associated service fibre and hardware, as schematically shown in Fig. 1b. Here, each user generates their local frequency comb and transmits one comb line to Charlie for interference, the outcome of which feeds into a photon-counting proportional-integral-derivative (PID) controller that allows rapid reading out and zeroing the laser frequency difference as well as cancelling the fibre fluctuation. Importantly, our solution brings neither performance degradation nor loss of practicality thanks partly to its choice of proven ingredients including ultra-stable lasers[30,33], optical frequency combs[31], and dual-wavelength stabilisation[28,31].

## Results

Our setup (Fig. 2) does not need service fibre and has an identical fibre configuration as MDI-QKD[13]. Alice and Bob are connected to Charlie from opposite directions via their segments of the quantum link. The quantum link is made of spools of ultra-low loss fibres with an average loss coefficient of 0.168 dB km$^{-1}$. For detailed information on the fibre properties, refer to Supplementary Table 1.

Each user owns an independent continuous-wave laser that features a sub-Hz short-term linewidth and allows adjustment of its optical frequency at 1 mHz step size. See Methods for the detailed information on the lasers. Passing through a phase modulator (PM), the laser light is modulated to generate a frequency comb (Fig. 2a) with a precise spacing of 25 GHz thanks to the microwave frequency driver referenced to a Rubidium frequency standard. For 50 GHz dense-wavelength-division-multiplexing (DWDM) compatibility, two comb lines of $\lambda_q = 1549.72$ nm and $\lambda_c = 1550.52$ nm (ITU Channels 34.5 and 33.5) are selected and filtered into two separate paths. The continuous-wave $\lambda_c$ signal is used to establish coherence with the other user, and we refer to it as the 'channel reference'. The $\lambda_q$ signal passes through the encoder box (Fig. 2b), which carves the continuous-wave input into

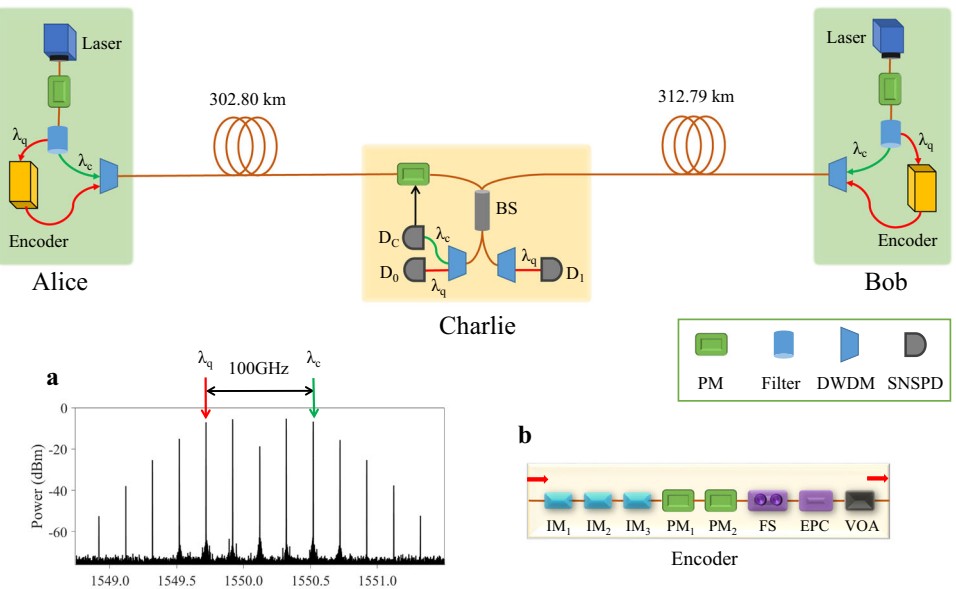

**Fig. 2 | Experimental setup.** Alice (Bob) owns an independent ultrastable laser, the signal of which is modulated by a phase modulator (PM) to produce a frequency comb of 25 GHz spacing. Two comb lines separated by 100 GHz are chosen for quantum signal encoding ($\lambda_q$) and channel stabilisation ($\lambda_c$), respectively. Charlie contains a receiving 50/50 beam splitter to interfere the incoming signals. The $\lambda_q$ photons are registered by $D_0$ and $D_1$ and the $\lambda_c$ photons by $D_c$. $D_c$'s count rate is used as error signal to the fast PID controller that cancels the twin-field phase fluctuation of the $\lambda_c$ signals. **a** Optical frequency comb measured after the PM; **b** Encoder box. DWDM dense wavelength-division multiplexing; EPC electrically driven polarisation controller; FS fibre stretcher; IM intensity modulator; VOA variable optical attenuator.

a train of pulses of 300 ps width at an interval of 1 ns. The encoder's description is provided in Supplementary Note 1. We apply blockwise modulation for every 200 pulses. The first 95 pulses of each block do not receive further modulation and are used to sense the phase of the quantum channel. They are referred to as the 'quantum reference'. The last 100 pulses are modulated independently according to TF-QKD protocol's requirement. As these quantum pulses can be considerably weaker than the quantum reference, the encoder extinguishes the 5 pulses in between to create an empty buffer to prevent inter-group contamination. Overall, our setup has an effective QKD clock rate of 500 MHz. After the encoder box, the $\lambda_q$ light is recombined with the channel reference ($\lambda_c$) into the quantum channel segment using a DWDM filter. In addition to setting the approximate photon fluxes, both wavelengths paths have separate polarisation controls for pre-compensation of polarisation rotation by the quantum link.

After travelling through their respective quantum link segments, Alice and Bob's light enters Charlie's 50/50 interfering beam splitter with an identical polarisation and matched intensities. The interference outcome is spectrally de-multiplexed before detection by three superconducting nanowire single photon detectors (SNSPD's), with $D_0$ and $D_1$ assigned to the $\lambda_q$ signals and $D_c$ for the $\lambda_c$ channel reference. Charlie contains a PM in one of his input arms for fast phase feedback control. Charlie's components losses and detector performance is summarised in Supplementary Tables 2 and 3.

We describe briefly below, and provide more information in Supplementary Note 2, on how our setup is stabilised. For all different fibre lengths, we adjust the channel references' intensities to have a maximal interference visibility and maintain an average count rate of approximately 13 MHz at $D_c$. This count rate allows 200 kHz sampling with acceptable noise by a field-programmable-gated-array (FPGA) PID controller (Supplementary Fig. 1) to process and generate compensation voltages to Charlie's PM, locking the differential phase to $\pi/2$ between the channel references. This locking process cancels simultaneously both instability terms in Eq. (1), i.e., the laser frequency difference and fibre fluctuation. As described later, the FPGA controller allows real-time readout of the laser frequency difference and can thus

apply feedback to one user's laser to prevent the laser drifting out of the PID's correction bandwidth. Due to coherence among comb lines, the phase instability by frequency difference for the $\lambda_q$ signals is reduced by a factor of $\frac{|\lambda_q - \lambda_c|}{\lambda_q}$, similarly to that by the fibre fluctuation[28]. The phase drift by this residual difference can then be corrected for through a second stage compensation[28] which uses the interference result of the quantum reference to act on a fibre stretcher in Alice's encoder at a rate of 50–100 Hz. Note that use of coherent combs enables the setup to support asymmetric channels, which can substantially ease fibre provision during installation.

To evaluate the effectiveness of our open fibre scheme, we analyse the compensation signal that is applied to the Charlie's PM in response to the lasers' frequency difference ($\Delta\nu$) and fibre fluctuation of the quantum channel. Figure 3a, b show the histograms of compensation signals integrated over 10 ms intervals for three different frequency offsets. For clarity, the PM compensation signal is converted to angular frequency with unit of $2\pi$ s$^{-1}$ or Hz. For a 10 m quantum channel (Fig. 3a), the PID compensates mostly the laser frequency difference and we therefore observe a histogram of sharp distributions with its compensating signal tracking exactly the laser frequency difference. After installing 615.6 km fibre (Fig. 3b), each histogram becomes considerably broadened because the PID has now to work on also rapid and random fibre fluctuation. However, the peak of each histogram remains at the same angular frequency, as the random fibre fluctuation does not produce an instant net drift. We can then determine the laser frequency drift with high accuracy by increasing the readout time to just 100 ms, as shown in Fig. 3c, where the compensation rate follows strictly the frequency offset. This fast readout enables real-time compensation for laser frequency drift, which is indispensable for long-term operation of TF-QKD.

Figure 3d shows the interference results of the channel references at Charlie's interfering 50/50 beam-splitter after travelling the 615.6 km quantum channel. With the PID switched off, the measured optical power oscillates violently between constructive and destructive extremes due to fast fibre fluctuation. We extract an average fringe visibility of 97.8 %, which is noticeably worse than the value of 99.0 %

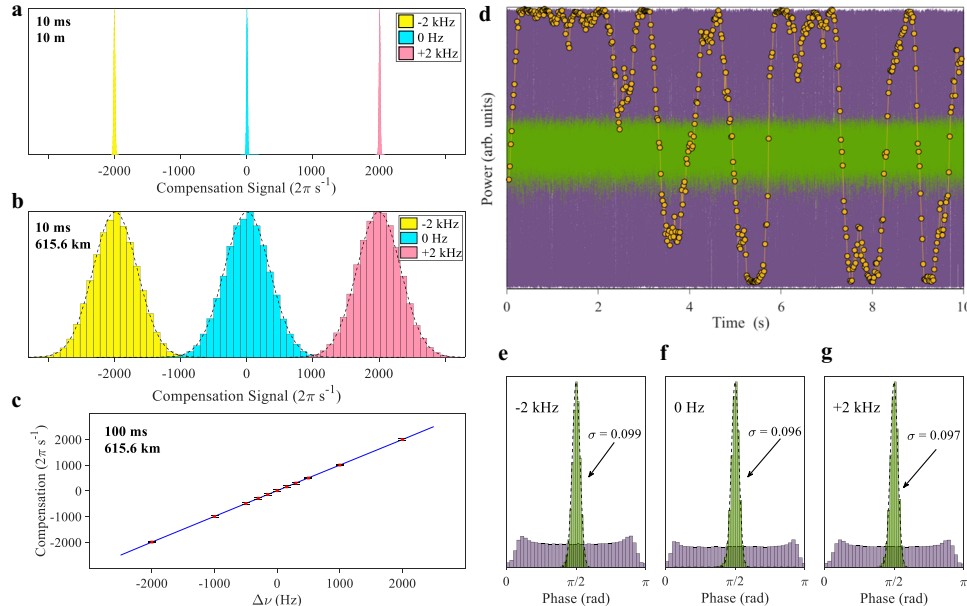

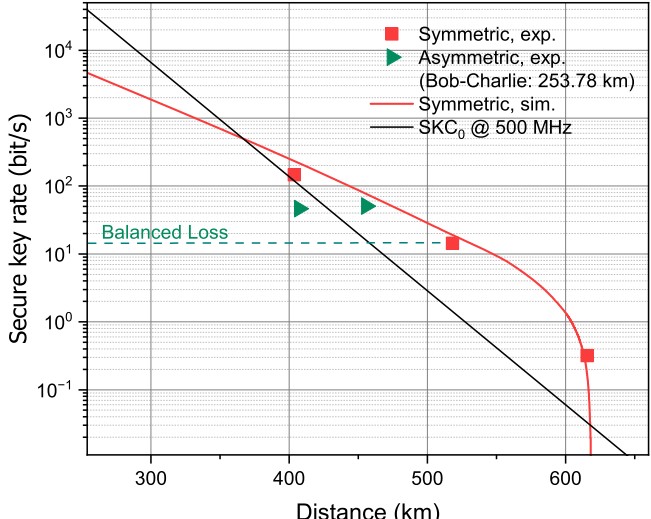

**Fig. 3 | Open quantum channel stabilisation.** Alice and Bob's lasers are fully independent, and their frequency difference is adjustable by offsetting Alice's laser to a high-fineness cavity. Except a, all data in this figure were measured with the 615.6 km quantum fibre. **a** Histograms of phase compensation signals integrated over 10 ms time intervals for a 10 m quantum link; **b** same as a but with 615.6 km quantum link; **c** compensation signal angular frequency as a function of the laser frequency offset ($\Delta\nu$). Error bars represent standard deviation. The blue line is a guide to the eye and has a slope of 1. **d** Optical output power as a function of time measured at one output of Charlie's 50/50 interfering beam-splitter; Purple (green): channel reference or the $\lambda_c$ signal when the FPGA PID controller is turned off (on); Orange: slowed drift of the $\lambda_q$ signal. **e**–**g** The phase angle distributions of the channel reference before (purple) and after (green) the simplex channel stabilisation, measured for laser frequency offsets of −2, 0 and 2 kHz, respectively. Symbol $\sigma$ represents standard deviation. Data in panels **d**–**g** were measured with a power metre.

obtained with a single laser serving both the users, illustrating a small penalty of using truly independent lasers. We extract a standard deviation of 1.07 kHz for the fibre drift, which is comparable to those reported in the literature[12,28,30,32]. After enabling the PID control, the interference output is narrowed down to a tight band (green). We extract the phase angles and plot their distributions as shown in Fig. 3e–g. We obtain almost identical standard deviations of 0.099, 0.096, and 0.097 rad for different frequency offsets of −2, 0, and +2 kHz, suggesting the PID control is tolerant to at least 2 kHz detuning. A phase deviation of 0.1 rad will cause about 0.5% drop in the interference visibility, which is acceptable for TF-QKD. After the PID control on the channel reference, the phase drift of the $\lambda_q$ signals is drastically slowed down, as shown by orange circles in Fig. 3d. The standard deviation of this drift rate is 0.72 Hz, which is 1500 times slower than the value for the unstabilised channel. We measure an interference fringe visibility of be 96.8 %, which will cause 1.6% floor to the QBER in the phase basis. For detailed information on how the visibility values were extracted, see Supplementary Note 4.

With the simplex channel stabilisation, we performed two sets of TF-QKD experiments using the SNS protocol[17,39,40]. In the first set, the users' channel losses to Charlie are strictly matched while their fibre lengths may differ by 10 km, which was introduced for balancing the loss by Charlie's PM. We ran the SNS protocol for three distances of 403.73, 518.16, and 615.59 km, with $2.025 \times 10^{12}$, $2.475 \times 10^{12}$, and $1.418 \times 10^{13}$ total pulses sent respectively. We take the actively odd-parity pairing (AOPP) method for data post-processing, which can efficiently reduce the bit-flip error rate of the raw key and have a higher probability for pairing success than the random 'two-way classical communication' method[39], and use the zigzag approach proposed in ref. [40] in the secure key rate (SKR) calculation. The detailed experimental parameters and results can be found in Supplementary Tables 4–6. In Fig. 4, we present our experimental results (red

**Fig. 4 | Secure key rate (SKR) simulations and results.** The AOPP-SNS TF-QKD with finite size effects was implemented in the experiments. Two sets of experimental data are included. Symmetric case (squares): The users' losses to Charlie are strictly matched while their fibre lengths may differ by 10 km; Asymmetric case (triangles): Bob's fibre length is fixed at 253.78 km. The green dashed line indicates the expected SKR when the asymmetry to Bob's 253.78 km fibre is treated by just adding attenuation. A fibre attenuation coefficient of 0.168 dB km$^{-1}$ is adopted for calculating the SKR simulation (red line) and the absolute repeaterless SKC$_0$ bound (black line) for an ideal point-point QKD setup operating at 500 MHz.

square) in terms of SKR versus distance together with the simulation curve (red line). We include also the absolute repeaterless Pirandola–Laurenza–Ottaviani–Banchi bound[9] (black line), i.e., SKC$_0$, which represents the fundamentally maximum rate that an

**Table 1 | A selection of recent long-haul TF-QKD experiments and comparison with this work**

| Experiment | Quantum/ service fibre | Frequency dissemination | Phase compensation | Number of wave-lengths | Inter- wavelength Coherence | Check-basis QBER | Bit-flip QBER |
|---|---|---|---|---|---|---|---|
| Wang et al.[32], 2022 | 833.8 km/833.8 km | Homodyne OPLL | Active | 1 | n/a | n/a | 3.79% |
| Clivati et al.[31], 2022 | 206 km/206 km | Heterodyne OPLL | Active, partial | 2 | Yes | n/a | n/a |
| Chen et al.[33], 2022 | 658.7 km/500 km | Time-freq. metrology | Post- selection | 1 | n/a | ~5.0% | 2.12% |
| Pittaluga et al.[28], 2021 | 605.2 km/611.4 km | Heterodyne OPLL | Active | 2 | No | 5.41% | 3.65% |
| This work | 615.6 km/ not needed | Not needed | Active | 2 | Yes | 4.75% | 1.97% |

All previous long-haul setups use optical frequency dissemination technologies, including homodyne/heterodyne optical phase locked loop (OPLL)[22,23,28,31,32], time-frequency metrology[24,27,33] and optical injection locking[26,29], to synchronise the users' lasers with the reference optical frequency delivered via a service fibre that is as long as the quantum fibre (see Fig. 1a). The subsequent twin-field phase can be either actively stabilised[23,28,31,32] or reconciled through post-selection at the data processing stage[26,30,33], though the latter approach does not support NPP-TF-QKD protocols. Among active schemes, using a second wavelength[28,31] can suppress double Rayleigh scattering noise and increase the stabilisation bandwidth, while further introduction of inter-wavelength coherence[31] enables support for asymmetric fibre channels. For comparing SNS-TF-QKD implementations, both the check (X-basis) and data bit-flip (Z-basis) quantum bit error rates (QBERs) are listed.

ideal point-point QKD could achieve and can only be overcome by a repeater-like setup. With finite-size effects being taken into consideration, we obtain SKR's of 146.7, 14.38 and 0.32 bit/s for 403.73, 518.16, and 615.59 km, respectively. They all overcome the $SKC_0$ bound, confirming the repeater-like behaviour of our setup. At 615.6 km, the SKR is 9.70 times above $SKC_0$.

To further demonstrate the robustness of our open channel scheme, we explore the capability of our setup supporting asymmetric fibre links. With Bob's fibre fixed at 253.78 km, we ran experiments over two different distances of 201.87 and 153.45 km between Alice and Charlie, representing a link asymmetry of 51.91 and 100.33 km, respectively. For fair assessment, we use an identical parameter set and compensate the extra loss asymmetry by adding 8.13 dB attenuation to the 153.45 km link. During optimisation of the parameter set, we apply mathematical constraint that need to be satisfied for the security of asymmetric SNS protocol[41] and introduce a new constraint for the intensities of both decoy states to guarantee a high interference visibility at Charlie. The experimental results are shown in Fig. 4 (green triangles). We extract respective finite-size SKR's of 50.75 and 46.30 bit/s for the asymmetries of 51.91 km and 100.33 km, with a minor deterioration by extra 48 km asymmetry. This result improves considerably over the current asymmetry record of 22 km[31], while most setups had to keep channels strictly matched down to metres[22,23,28,32]. Additionally, our result illustrates also the importance of parameter optimisation for asymmetric links, as the obtained SKR's are thrice higher than the rate (green dashed line) expected if we just add attenuation to balance the fibre disparity.

We compare our open scheme with recent experiments that adopted the closed MZI configuration. As SKR's and distances are directly affected by the detector performance and/or the clock frequency, the relevant parameter to compare here is the quantum bit error rate (QBER) arising from the interfering twin-field signals that went through TF-QKD's phase randomisation process. Based on this criteria, we list in Table 1 the X-basis QBER for experiments that adopted also the SNS protocol. Within the margin of error, our system gives even a slightly better QBER of 4.75 % than the other setups[28,33], showing no performance degradation from using truly independent lasers between an open quantum channel.

## Discussion

With the open-channel stabilisation technique we are able to achieve a simple and robust TF-QKD setup that can drastically ease fibre provision and route planning for future deployment. The technique could be adapted to enable free-space quantum experiments involving single-photon interference between remote light sources, including free-space TF-QKD. Moreover, its demonstrated frequency tolerance makes it possible to use less-

demanding lasers, e.g., these lasers[42,43] that reference to absolute atomic or molecular transitions and feature typically 1 kHz linewidth, stimulating further simplification in TF-QKD setups. We believe our technique is applicable to phase-based quantum applications in general, including quantum repeaters[11], single photon entanglement distribution[44], and quantum internet[45].

## Methods
### Ultra-stable lasers
The ultra-stable lasers were manufactured by MenloSystems (Model: ORS-Cubic). Each laser emits at a wavelength of 1550.12 nm and features a sub-Hz short-term linewidth thanks to its use of Pound–Drever–Hall (PDH) technique for locking to a cavity with a fineness of 250,000 and a free spectral range (FSR) of 3 GHz. In the PDH locking path there is an extra phase modulator that is driven at a base frequency of 300 MHz and allows fine adjustment of the laser frequency with a step size of 1 mHz. We measured the two lasers to have respective frequency drift rates of 110.4 mHz s$^{-1}$ and 92.5 mHz s$^{-1}$ in the same direction, and their differential frequency drifts about 1500 Hz per day. During initial characterisation of the TF-QKD setup, the frequency difference ($\Delta\nu$) between the lasers was monitored through their beating note recorded by a photodiode and can be precisely set via adjusting the modulation frequency to the PM in one laser. This laser frequency drift can easily be corrected for by our photon-counting-based FPGA PID controller, so the offset monitoring by the photodiode is unnecessary during TF-QKD experiments. Our laser frequency feedback will work even when they are installed in separated locations.

### Coherent frequency comb
We generate the electro-optic frequency comb by passing the continuous-wave laser through a phase modulator that is driven by a stable microwave source with its power carefully set. Driving at 25 GHz, we obtain a total of 13 comb lines between 1548.9 and 1551.3 nm as shown in Fig. 2a. The comb lines of 1549.72 nm ($\lambda_q$) and 1550.52 nm ($\lambda_c$) are selected as the quantum signal and the channel reference respectively. Their spacing of 0.8 nm (100 GHz) is compatible with the ITU G694.1 standard DWDM grid and allows convenient spectral filtering and wavelength routing. After spectral filtering, each comb line has an output power of ≥300 µW and channel isolation of >55 dB, both sufficient for TF-QKD encoding.

### Protocol
In this experiment, we adopt a 4-intensity SNS-TF-QKD protocol with actively odd-parity pairing (AOPP)[39] for the data post-processing, with finite-size effects being taken into account. We describe the theory of asymmetric protocol[41] and mention that it also applies to the symmetric case when Alice and Bob have an identical loss to Charlie and use the same values for their source parameters.

In this protocol, Alice and Bob repeat the following process $N_{\text{tot}}$ times to obtain a string of binary bits. In each time window, Alice (Bob) randomly decides whether it is a decoy window with probability $p_{Ax}$ ($p_{Bx}$) or a signal window with probability $1 - p_{Ax}$ ($1 - p_{Bx}$). If it is a signal window, Alice (Bob) randomly prepares a phase-randomised weak coherent state (WCS) with intensity $\mu_{Az}$ ($\mu_{Bz}$) and decides whether to send it or not, with probabilities $\epsilon_A$ ($\epsilon_B$) and $1 - \epsilon_A$ ($1 - \epsilon_B$), respectively. For the decisions of not-sending in the signal windows, Alice (Bob) denotes them as bit 0 (1), and for the decisions of sending, Alice (Bob) denotes them as bit 1 (0). If a decoy window is chosen, Alice (Bob) randomly prepares phase-randomised WCS with intensities $\mu_{A0}$ ($\mu_{B0}$), $\mu_{A1}$ ($\mu_{B1}$) or $\mu_{A2}$ ($\mu_{B2}$) with respective probabilities $1 - p_{A1} - p_{A2}$ ($1 - p_{B1} - p_{B2}$), $p_{A1}$ ($p_{B1}$) and $p_{A2}$ ($p_{B2}$). As was proven in ref. [41], in order to main the security of the protocol, the asymmetric source parameters should satisfy the following mathematical constraint

$$\frac{\mu_{A1}}{\mu_{B1}} = \frac{\epsilon_A(1 - \epsilon_B)\mu_{Az}e^{-\mu_{Az}}}{\epsilon_B(1 - \epsilon_A)\mu_{Bz}e^{-\mu_{Bz}}}. \qquad (2)$$

Note that this requirement is automatically met for the symmetric protocol, and its purpose is to guarantee the users' raw key free from systematic bias. After the preparation stage, Alice and Bob send their pulses to the middle untrusted node, Charlie, who performs interference measurements and announces publicly which detector clicks. The detection events which one and only one detector clicks are taken as effective events. For time windows determined by both Alice and Bob to be a signal window, which are labelled as Z windows, Alice and Bob get two $n_t$ bits of raw key strings comprised by the corresponding bits from effective events. The bit-flip error rate of these two strings is denoted as $E_z$.

Events in time windows determined by both Alice and Bob to be a decoy window, which are labelled as X windows, are used to perform the security analysis. In X windows where Alice and Bob choose intensities $\mu_{A1}$ and $\mu_{B1}$, respectively, the phase information of their WCSs would be publicly announced and post-selected based on the following criteria

$$|\theta_{A1} - \theta_{B1}| \le \frac{2\pi}{M} \quad \text{or} \quad |\theta_{A1} - \theta_{B1} - \pi| \le \frac{2\pi}{M}, \qquad (3)$$

where $\theta_{A1}$ and $\theta_{B1}$ are the private phases of Alice's and Bob's pulses respectively and $M$ is the number of phase slices.

The AOPP method is used to reduce the errors of raw key strings before the error correction and privacy amplification processes. In AOPP, Bob first actively pairs his bits 0 with bits 1 of the raw key string and announces the pairing information to Alice. Alice performs the same pairing accordingly and they then compare the parity of pairs. They discard both bits in the pair if the announced parities are different and keep the first bit of the pairs if the parities are the same. The users now use the remaining bits to form a new shorter string $n'_t$ with dramatically reduced bit-flip error rate $E'_z$, from which they will extract the final key.

Next we briefly show how to extract the information of single-photon states, which constitutes the final key rate formula, from X-window events, through decoy-state analysis. We denote the counting rate of sources $\kappa\zeta$ in X windows by $S_{\kappa\zeta}$, which is the ratio of the number of corresponding effective events to the number of respective pulses sent out by Alice and Bob. These values can be measured in the experiment. Note that a statistical fluctuation analysis should be considered here as part of the finite-size effects, with more details being presented in ref. [46]. Then we use the decoy-state method to deduce the counting rate of single-photon states which either Alice or Bob actually sends out a single photon from WCSs, which are[41],

$$\langle \underline{y_{10}} \rangle = \frac{\mu_{A2}^2 e^{\mu_{A1}} \langle \underline{S_{\mu_{A1}\mu_{B0}}} \rangle - \mu_{A1}^2 e^{\mu_{A2}} \langle \overline{S_{\mu_{A2}\mu_{B0}}} \rangle - (\mu_{A2}^2 - \mu_{A1}^2)\langle \overline{S_{\mu_{A0}\mu_{B0}}} \rangle}{\mu_{A2}\mu_{A1}(\mu_{A2} - \mu_{A1})}, \qquad (4)$$

$$\langle \underline{y_{01}} \rangle = \frac{\mu_{B2}^2 e^{\mu_{B1}} \langle \underline{S_{\mu_{A0}\mu_{B1}}} \rangle - \mu_{B1}^2 e^{\mu_{B2}} \langle \overline{S_{\mu_{A0}\mu_{B2}}} \rangle - (\mu_{B2}^2 - \mu_{B1}^2)\langle \overline{S_{\mu_{A0}\mu_{B0}}} \rangle}{\mu_{B2}\mu_{B1}(\mu_{B2} - \mu_{B1})}, \qquad (5)$$

respectively, where the notations $\langle \underline{\cdot} \rangle$ and $\langle \overline{\cdot} \rangle$ denote the lower and the upper bound of the corresponding expected values, respectively, with a composable definition of security and the Chernoff bound being applied. Their detailed explanations and expressions can be found in the ref. [47]. Then the lower bound of the expected value of the counting rate of untagged bits, which is the number of bits generated through effective events when the users actually send out single-photon states in Z windows, is given by[41]

$$\langle \underline{y_1} \rangle = \frac{\mu_{A1}}{\mu_{A1} + \mu_{B1}} \langle \underline{y_{10}} \rangle + \frac{\mu_{B1}}{\mu_{A1} + \mu_{B1}} \langle \underline{y_{01}} \rangle \qquad (6)$$

and the lower bound of the expected value of the number of untagged bits is

$$\langle \underline{n_1} \rangle = N_{ZZ}\left[ \epsilon_A(1 - \epsilon_B)\mu_{Az}e^{-\mu_{Az}} \langle \underline{y_{10}} \rangle + \epsilon_B(1 - \epsilon_A)\mu_{Bz}e^{-\mu_{Bz}} \langle \underline{y_{01}} \rangle \right], \qquad (7)$$

where $N_{ZZ} = N_{\text{tot}}(1 - p_{Ax})(1 - p_{Bx})$ is the number of pulses Alice and Bob both choose signal windows. According to ref. [41], so long as Eq. (2) is satisfied, the phase-flip error rate of untagged bits in Z windows can be calculated from the bit-flip error rate of untagged bits in X windows, which is written as

$$\langle \overline{e_1^{ph}} \rangle = \frac{\langle \overline{T_{XX}} \rangle - 1/2e^{-\mu_{A1} - \mu_{B1}} \langle \underline{S_{\mu_{A0}\mu_{B0}}} \rangle}{e^{-\mu_{A1} - \mu_{B1}}(\mu_{A1} + \mu_{B1})\langle \underline{y_1} \rangle} \qquad (8)$$

where $T_{XX}$ is the ratio of the number of corresponding error events over the number of total pulses with intensities $\mu_{A1}$ and $\mu_{B1}$ sent out in X windows.

The secret key rate (SKR) of AOPP with finite-size effects is given by

$$R = \frac{1}{N_{\text{tot}}}\{n'_1[1 - h(e_1'^{ph})] - fn'_t h(E'_z) - \Delta\}, \qquad (9)$$

where $h(x) = -x\log_2 x - (1 - x)\log_2(1 - x)$ is the binary Shannon entropy. $f = 1.1$ is the error correction efficiency factor. $\Delta = 2\log_2(2/\epsilon_{\text{cor}}) + 4\log_2(1/\sqrt{2}\epsilon_{\text{PA}}\hat{\epsilon})$ is the finite-size correction term, with $\epsilon_{\text{cor}} = 10^{-10}$, $\epsilon_{\text{PA}} = 10^{-10}$ and $\hat{\epsilon} = 10^{-10}$ being the failure probabilities for error correction, privacy amplification and the coefficient of the smoothing parameter, respectively. $n'_1$ and $e_1'^{ph}$ are the number of untagged bits and their phase-flip error rate, respectively, after AOPP process. Here we adopt a zigzag approach proposed in ref. [40] in order to obtain higher key rates and take all the finite-key effects efficiently, the same as the calculation method applied in experimental reference[29,30,33]. For simplicity, we do not list the calculation processes here, with all details can be found in the cited papers.

## Data availability
The data that support the plots within this paper are deposited on Zenodo[48].

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

## Acknowledgements

We thank X. B. Wang for helpful discussions on the SNS-AOPP TF-QKD protocol. This work was supported by the National Natural Science Foundation of China under grants 62105034 (L.Z.) and 62250710162 (Z. Y.).

## Author contributions

L.Z. and J.L. developed the experimental setup, performed the experiments, collected and analysed the data. Y.J. performed the simulation. Z.Y. supervised the project, and wrote the manuscript with input from all authors.

## Competing interests

The authors declare no competing interests.
