## [Peer Review File · Nature Communications]

Twin-field quantum key distribution without optical frequency disseminationREVIEWER COMMENTS

Reviewer #1 (Remarks to the Author):

Comments to the authors

In the manuscript titled “Twin-field quantum key distribution without optical frequency dissemination”, the authors introduce a novel technique that allows them to experimentally implement a variant of TF-QKD without resorting to service fibres between the users and the intermediate node.

Most current implementations of TF-QKD protocols use optical frequency dissemination technologies over service fibres in order to remove the phase drifts in the quantum signals of the two parties. Phase differences between the signals of Alice and Bob are extremely detrimental for the protocol’s key rate and must be corrected as much as possible. The use of service fibres however increases the complexity and cost of the setup, are not compatible with active optical switching and prevent applying TF-QKD in free-space links.

The solution proposed in this manuscript removes the need of service fibres or Sagnac loops. Each user has an independent ultra-stable laser and uses it to prepare a frequency comb with a total of 13 lines. One line is selected to encode the quantum signal used by the protocol and another line to correct the phase drift. The interference of the latter comb line at the intermediate node is used to check and correct for phase drifts between the two quantum signals of the users, thereby correcting most of the phase drift. The residual slower drift is corrected by checking the interference of a portion of the quantum signal encoded on the former comb line.

In my opinion, the solution proposed to correct phase drifts using signals with different wavelengths resembles the solution proposed in Ref [27] of the manuscript. There, the intermediate node sends a dual-band signal to both users through service fibres and then each users encodes their quantum signal on one of the two wavelengths. Despite the resemblance, the protocol in Ref [27] uses service fibres that, together with the fibres used for the quantum signal, form a large Mach-Zehnder interferometer (MZI) and thus carries the drawbacks mentioned above about complexity, cost, and incompatibility. Hence, the real innovation brought by this manuscript is the use of independent lasers by each user and the use of frequency combs, which can drastically simplify the implementation and deployment of TF-QKD. The authors furthermore demonstrate the feasibility of their solution by running a TF-QKD protocol and computing its key rate with finite-size effects. Importantly, with Table 1 they show that, despite the use of independent lasers, their error rates are lower than that of previous works that used MZI configurations. At the same time, their overall key rates are comparable to those of the best state-of-the-art in the field (Fig 4). Overall, I consider these results quite impressive.

For the above reasons, although a somehow similar solution was already proposed in Ref [27], I consider this manuscript innovative and with noteworthy results such that it deserves to be considered for publication in Nature Communications. Nevertheless, I have few comments that I would like the authors to address before accepting the paper.

Comments

1. The Table 1 where the QBER comparison is performed is really useful. However, the QBERs of Refs. [30] and [31] are not reported. Would it be possible to obtain those from the authors of the corresponding papers?
2. The authors argue that the best figure of merit to compare their work with other works that display different setups is the QBER in the X basis. Although I can agree with this statement, I would also be interested in comparing the QBERs in the Z basis, since that is the quantity that describes how correlated are the key bits of Alice and Bob, correct? I consider this a very important figure to have in another table, similar to Table 1.
3. Again regarding the comparison with other works, I believe that a comparison with the TF-QKD implementations that are based on Sagnac loops is missing. Would it be possible to compare some performance measure in this case? If not, why? Are they too underperforming with respect to the experiment of this manuscript? If so, it would be useful to mention this. What are other advantages of the authors’ approach compared to the approach with Sagnac loops?
4. The formulas in Eq (1) and in line 128 come without a proper explanation or referencing. Please refer to other literature where the formulas are derived or derive them in the paper.

5. When the authors say “The last 100 pulses are modulated according to TF-QKD protocol’s requirement” I understand that each of the 100 pulses is modulated independently of the others. And this is how it should be. If this is the case, please specify that the pulses are modulated independently of each other. In case the 100 pulses are instead modulated all in the same way, I think that this would lead to a security loophole since Eve would know that the parties send trains of 100 identical pulses.

Reviewer #2 (Remarks to the Author):

Lai Zhou et al. experimentally demonstrated the possibility of twin-field quantum key distribution (TF-QKD) without optical frequency dissemination. The advantages of TF-QKD have been shown in several recent experiments, and now new methods and technologies are needed to improve the practicality of TF-QKD. This paper brought us a fresh solution. As far as I know, this is the first TF-QKD experiment without the need of service fibers, and the ingenious idea with electro-optic frequency comb made the performance of TF-QKD almost unchanged. This work is very solid, and will be of great interest of a wide range of scientists and engineers, especially for quantum communication. I think this work deserves publication in Nature Communications.

However, I have some comments, reported below, mainly about technique details and expressions, to make it possible to reproduce them.

1. About experimental setup, I understand the experiment setup of this through Fig.2 and Fig.S1.

- 1) The line separation of quantum signal and channel stabilization is 100 GHz, there are still three lines between the used two lines. Please give details of the filter to get 55 dB isolation. And also, the isolation of DWDMs in Alice's and Charlie's sites are important to get small scattered noise.
- 2) There are two EPCs in Alice's (or Bob's) node, please explain the reasons why use them. In my opinion, one EPC might be enough to compensate the polarization drift of fiber channel.
- 3) In Fig.S1, three signals are needed to be transmitted from Charlie to Alice, also to Bob, to active feedback the laser frequency, FS, two EPCs. With long distance, the transmission of signals need spend time, it would reduce the feedback bandwidth of the whole system.
- 4) Please explain the roles of FS (Alice) and PM (Charlie), and the differences of them. The feedback rate to drive Charlie's PM is 200 kHz, why chose such a value of rate. If the rate is lower, it might be possible to replace Charlie's PM with FS to reduce the insertion loss.
- 5) The values of intensity of quantum reference and channel reference are missing, please offer them. These values are useful for estimation of scattered noise and feedback rate.
- 6) Line 103~104, 'the encoder extinguishes the 5 pulses in between to create an empty buffer to prevent inter-group contamination', please explain the reasons to choose 5 pluses. In ref.27 with dual-band stabilization, for the quantum wavelength, the even-numbered pulses are quantum signals, while the odd-numbered pulses that are referred as 'dim reference' pulses are used to track the phase drift of the quantum signals, there is no empty buffer.

2. About disadvantage of the idea with electro-optic frequency comb, the ingenious idea solve two severe drawbacks of previous TF-QKD systems, here I also want to point out that this idea still has two small 'drawbacks'. One is it complicates Charlie's setup and increases the insertion loss, though it simplifies Alice's and Bob's setups and removes service fiber channels. The other is it increase the numbers of single photon detectors, there are two more detectors were employed in Fig. S1. Please give some discussion about these two small 'drawbacks'.

3. About expression of some sentences

- 1) First paragraph in the second page (line 33), 'but extracts information from single photon interference rather than two-photon coincidence', most information is extracted from single photon, but we still can extract information from multi-photon states. Please give a more strict expression.
- 2) Equation (1), 's light speed in the fibre, and ΔL the length difference between the users' fibres to Charlie', I think the refractive index of the fiber is also one key factor that introduces differential phase. One better expression might be $\Delta(nL)$, the authors can see equation (1) in ref. 31.
- 3) The second paragraph of supplementary information, 'IM_1 is modulated for pulse carving, IM_2 by a waveform signal for preparing pulses of different intensities', and 'The encoder is able to achieve >40 dB extinction ratio between the signal (μ_Z) and vacuum (μ_0) states.' I think there might be some misleading expression about such high extinction ratio with only one intensity modulator. Please carefully check it.
- 4) In 5th page of supplementary information, the line numbers (99~112) overlap with TABLE S2, please correct the box of TABLE S2 in the revised version.

Point-to-point response

We provide point-to-point response to the reviewers's comments below. Reviewers' comments are shown in *Italic*, and our replies in **bold**.

Reviewer #1.

In the manuscript titled 'Twin-field quantum key distribution without optical frequency dissemination', the authors introduce a novel technique that allows them to experimentally implement a variant of TF-QKD without resorting to service fibres between the users and the intermediate node.

Most current implementations of TF-QKD protocols use optical frequency dissemination technologies over service fibres in order to remove the phase drifts in the quantum signals of the two parties. Phase differences between the signals of Alice and Bob are extremely detrimental for the protocol's key rate and must be corrected as much as possible. The use of service fibres however increases the complexity and cost of the setup, are not compatible with active optical switching and prevent applying TF-QKD in free-space links.

The solution proposed in this manuscript removes the need of service fibres or Sagnac loops. Each user has an independent ultra-stable laser and uses it to prepare a frequency comb with a total of 13 lines. One line is selected to encode the quantum signal used by the protocol and another line to correct the phase drift. The interference of the latter comb line at the intermediate node is used to check and correct for phase drifts between the two quantum signals of the users, thereby correcting most of the phase drift. The residual slower drift is corrected by checking the interference of a portion of the quantum signal encoded on the former comb line.

In my opinion, the solution proposed to correct phase drifts using signals with different wavelengths resembles the solution proposed in Ref [27] of the manuscript. There, the intermediate node sends a dual-band signal to both users through service fibres and then each users encodes their quantum signal on one of the two wavelengths. Despite the resemblance, the protocol in Ref [27] uses service fibres that, together with the fibres used for the quantum signal, form a large Mach-Zehnder interferometer (MZI) and thus carries the drawbacks mentioned above about complexity, cost, and incompatibility. Hence, the real innovation brought by this manuscript is the use of independent lasers by each user and the use of frequency combs, which can drastically simplify the implementation and deployment of TF-QKD. The authors furthermore demonstrate the feasibility of their solution by running a TF-QKD protocol and computing its key rate with finite-size effects. Importantly, with Table 1 they show that, despite the use of independent lasers, their error rates are lower than that of previous works that used MZI configurations. At the same time, their overall key rates are comparable to those of the best state-of-the-art in the field (Fig 4). Overall, I consider these results quite impressive.

For the above reasons, although a somehow similar solution was already proposed in Ref [27], I consider this manuscript innovative and with noteworthy results such that it deserves to be considered for publication in Nature Communications. Nevertheless, I have few comments that I would like the authors to address before accepting the paper.

We appreciate the reviewer's thorough understanding of the research subject and thank for their recognition of our manuscript as "innovative" and "noteworthy". We agree with the reviewer's assessment of our progress over Ref. [27] (new Ref. [28]) and are pleased that the reviewer's comment that our manuscript "deserves to be considered for publication in Nature Communications".

The Table 1 where the QBER comparison is performed is really useful. However, the QBERs of Refs. [30] and [31] are not reported. Would it be possible to obtain those from the authors of the corresponding papers?

Refs. [30,31] (new [31, 32]) implemented non phase post-selection (NPP) TF-QKD protocol, which does require the QBER in the check basis for security analysis and extracting secure keys unlike SNS-TF-QKD protocol we implemented. Ref [31] (new [32]) did not report this QBER. Nor did Ref. [24] (new [25]) which implemented also NPP-TF-QKD protocol using a Sagnac loop structure. Ref [30] (new [31]) demonstrated the phase stabilisation but did not implement a complete TF-QKD system. Therefore, we are not able to list the corresponding QBER for NPP-TF-QKD implementations.

NPP and SNS TF-QKD protocols often use different terminologies for the check basis. To avoid ambiguity, we refer to the X-Basis QBER as “Check-basis QBER” for SNS TF-QKD protocols in Table I. For further clarify, we have re-organised Table-I so that SNS-TF-QKD and NPP-TF-QKD implementations are clearly separated.

The authors argue that the best figure of merit to compare their work with other works that display different setups is the QBER in the X basis. Although I can agree with this statement, I would also be interested in comparing the QBERs in the Z basis, since that is the quantity that describes how correlated are the key bits of Alice and Bob, correct? I consider this a very important figure to have in another table, similar to Table 1.

We agree to the importance of QBERs in the Z-basis (SNS protocols) and have therefore added a new column in Table I. We refer to the Z-Basis QBER as “bit-flip QBER” in order to avoid confusion from NPP-TF-QKD protocols, which have different references to X and Z bases.

We note SNS protocol takes advantage of post pairing that substantially reduces the bit-flip error in the data basis using either ‘two-way classical communication (TWCC)’ or more efficient ‘actively odd-parity pairing (AOPP)’ method. As the data is extracted from photon arrival times, the bit-flip QBER is independent of the interference visibility between Alice and Bob’s laser signals and therefore can be reach excellent values. In contrast, NPP protocols distil their raw key from phase encoded pulses and its bit-flip error will subject to the interference visibility. Hence, comparing bit-flip QBER’s is a direct comparison for performance between different protocols.

Ref [30] (new 31) did not implement a complete TF-QKD and hence its data is not available.

Again regarding the comparison with other works, I believe that a comparison with the TF-QKD implementations that are based on Sagnac loops is missing. Would it be possible to compare some performance measure in this case? If not, why? Are they too underperforming with respect to the experiment of this manuscript? If so, it would be useful to mention this. What are other advantages of the authors’ approach compared to the approach with Sagnac loops?

We did not include Sagnac in comparison for two reasons. Firstly, Table I lists only long-haul TF-QKD implementations over more than 100 kilometers of fibres, while existing Sagnac setups used less than 10 km fibre. In fact, Sagnac’s long-haul capability has so far not been proven and could be obstructed by noise contamination due to counter-propagating signals of strong intensity disparity. Secondly, TF users in a Sagnac loop are subjected Trojan horse and phase-remapping attacks, because they uses Charlie’s transmitted pulses for quantum signal preparation.

With above consideration, we remain our preference not to include Sagnac implementations in Table 1. We believe existing discussion (lines 49-51) in the introduction is fair and ample for Sagnac implementations.

The formulas in Eq (1) and in line 128 come without a proper explanation or referencing. Please refer to other literature where the formulas are derived or derive them in the paper.

We derived Eq. (1) ourselves, and did not include it in the manuscript due to its simplicity. Before showing here, we first rewrite Eq. (1) into a new, more precise form according to Reviewer 2's recommendation:

$$\frac{d\phi}{dt} = 2\pi(\Delta\nu + \frac{\nu}{c} \frac{d\Delta(nL)}{dt}), \quad (1)$$

where c is the light speed in vacuum, and $\Delta(nL)$ represents the *optical* length difference between Alice and Bob's fibre channels. We further note that $\Delta\nu$ and $\Delta(nL)$ are both functions of time t and $\Delta\nu \ll \nu$. We start our derivation by assuming, without loss of generality, that

$$\phi(t) = 0. \quad (2)$$

Then, at $t + dt$, the differential phase can be written as

$$\phi(t + dt) = 2\pi\Delta\nu dt + \frac{2\pi\nu}{c} [\Delta(nL)_{t+dt} - \Delta(nL)_t], \quad (3)$$

where the first term represents the phase evolution due to the frequency difference, and the second term the phase difference from the change in the optical paths. Hence, the phase difference evolves as

$$\frac{d\phi}{dt} = \frac{\phi(t + dt) - \phi(t)}{dt} = 2\pi(\Delta\nu + \frac{\nu}{c} \frac{d\Delta(nL)}{dt}). \quad (4)$$

We reach Eq. (1). Note that our Eq. (1) has a different form from Eq. (4) used in original TF-QKD paper (Ref. [12]), M. Lucamarini *et al.*, Nature 414, 400 (2018). The primary difference is that the latter considered only the phase accumulation by the fibre and hence lacks the phase evolution term $2\pi\Delta\nu$.

Regarding line 128, we are very grateful to the reviewer's comment. The statement of line 128 was in fact scientifically inaccurate because the finite range of Charlie's PM does not allow correction of the frequency difference. We are glad to have the opportunity to rectify it. The revised sentence reads as "Due to coherence among comb lines, the **phase instability** by frequency difference for the λ_q signals is reduced by a factor of $\frac{|\lambda_q - \lambda_c|}{\lambda_q}$, **similarly to that by the fibre fluctuation [28].**" This revision addresses also the reviewer's comment on the formula.

When the authors say "The last 100 pulses are modulated according to TF-QKD protocol's requirement" I understand that each of the 100 pulses is modulated independently of the others. And this is how it should be. If this is the case, please specify that the pulses are modulated independently of each other. In case the 100 pulses are instead modulated all in the same way, I think that this would lead to a security loophole since Eve would know that the parties send trains of 100 identical pulses.

We thank the reviewer for raising this point. Precisely as the reviewer pointed out, each of the 100 pulses was modulated independently of the others. To improve clarity, we have revised the sentence to read as "The last 100 pulses are modulated **independently** according to TF-QKD protocol's requirement."

Reviewer #2.

Lai Zhou et al. experimentally demonstrated the possibility of twin-field quantum key distribution (TF-QKD) without optical frequency dissemination. The advantages of TF-QKD have been shown in several recent experiments, and now new methods and technologies are needed to improve the practicality of TF-QKD. This paper brought us a fresh solution. As far as I know, this is the first TF-QKD experiment without the need of service fibers, and the ingenious idea with electro-optic frequency comb made the performance of TF-QKD almost unchanged. This work is very solid, and will be of great interest of a wide range of scientists and engineers, especially for quantum communication. I think this work deserves publication in Nature Communications.

We thank the reviewer for their positive comments and recommendation to publish the work.

However, I have some comments, reported below, mainly about technique details and expressions, to make it possible to reproduce them.

About experimental setup, I understand the experiment setup of this through Fig.2 and Fig.S1.

1) The line separation of quantum signal and channel stabilization is 100 GHz, there are still three lines between the used two lines. Please give details of the filter to get 55 dB isolation. And also, the isolation of DWDMs in Alice's and Charlie's sites are important to get small scattered noise.

We used programmable filters (0.2 nm bandwidth, 500 dB/nm slope steepness) to first separate out the desired comb lines of λ_c (1550.52 nm) and λ_q (1549.72 nm). This filtering gives an isolation of > 40 dB from surrounding comb lines. Further isolation of >15 dB arises from the subsequent 50 GHz DWDM multiplexer that combines the quantum wavelength and the channel reference. The final isolation is greater than 55 dB.

Isolation of Charlie's DWDM was measured to be >68 dB between λ_q and λ_c . This information is now provided in the Caption of Table S3 and its text description in Section III of Supplementary Information.

2) There are two EPCs in Alice's (or Bob's) node, please explain the reasons why use them. In my opinion, one EPC might be enough to compensate the polarization drift of fiber channel.

Two EPC's are necessary in our setup. Due to the large spectral separation of 100 GHz between λ_q and λ_c and long fibre, wavelength dispersion prevents precise polarisation compensation for both wavelengths simultaneously when just a common EPC is used. Hence, we add a second EPC to correct for the polarisation difference.

3) In Fig.S1, three signals are needed to be transmitted from Charlie to Alice, also to Bob, to active feedback the laser frequency, FS, two EPCs. With long distance, the transmission of signals need spend time, it would reduce the feedback bandwidth of the whole system.

We thank the reviewer for raising this pertinent point.

Signal transmission over the maximal distance of 300 km fibre takes about 1.56 milliseconds. This latency will have negligible impact on the laser frequency and EPC feedback routines, which operate daily and every 100-200 milliseconds, respectively. The FS feedback routine runs considerably faster, but its feedback period of 10-20 milliseconds remains about 10 times larger than, and will therefore be sufficient to accommodate, the communication latency.

4) Please explain the roles of FS (Alice) and PM (Charlie), and the differences of them. The feedback rate to drive Charlie's PM is 200 kHz, why chose such a value of rate. If the rate is lower,

it might be possible to replace Charlie's PM with FS to reduce the insertion loss.

Charlie's PM is for fast phase compensation (λ_c , 200 kHz), and Alice's FS for compensating the slow, residual phase drift of the quantum signal (λ_q). We chose 200 kHz for Charlie's PM because it gave an optimal result in our experiment.

While it is possible to the requirement for PM feedback rate, we note that there already exist off-the-shelf fibre stretchers offering a large signal bandwidth of 250 kHz, see <https://www.idil-fibres-optiques.com/product/fiber-stretcher-2/>. Such FS could be viable replacement for the PM, and can further reduce Charlie's loss as the reviewer correctly anticipated. However, this potential improvement is out of the scope of the present investigation and nor does it affect our conclusion.

5) The values of intensity of quantum reference and channel reference are missing, please offer them. These values are useful for estimation of scattered noise and feedback rate.

We have added one sentence in the revised Supplementary Information, Section V, reading as **“The respective mean photon numbers for the quantum reference are 0.53, 3.02 and 13.1 photon/pulse for fibre lengths of 403.73, 518.16 and 615.59 km.”**

6) Line 103 104, ‘the encoder extinguishes the 5 pulses in between to create an empty buffer to prevent inter-group contamination’, please explain the reasons to choose 5 pluses. In ref.27 with dual-band stabilization, for the quantum wavelength, the even-numbered pulses are quantum signals, while the odd-numbered pulses that are referred as ‘dim reference’ pulses are used to track the phase drift of the quantum signals, there is no empty buffer.

Closely spaced signals cause a special type of quantum bit error due to finite time-resolution of single photon detectors. Its severity is amplified for a quantum pulse that is immediately transmitted immediately after a much stronger ‘reference’ pulse. Hence, we choose to sacrifice 5 time slots in order to minimise the QBER. Ref [28] chose to interleave weak quantum and stronger ‘quantum reference’ pulses and therefore their Z-basis (bit-flip) QBER is noticeably worse than ours, see Table I.

About disadvantage of the idea with electro-optic frequency comb, the ingenious idea solve two severe drawbacks of previous TF-QKD systems, here I also want to point out that this idea still has two small ‘drawbacks’. One is it complicates Charlie's setup and increases the insertion loss, though it simplifies Alice's and Bob's setups and removes service fiber channels. The other is it increase the numbers of single photon detectors, there are two more detectors were employed in Fig. S1. Please give some discussion about these two small ‘drawbacks’.

We respectfully point out that our Charlie's setup is no more complex than previous TF-QKD setup that adopts dual-band stabilisation technique (Ref. 28). As compared with single-band systems (Refs. 32,33), introduction of a second wavelength band allows for a greater control in phase compensation and faster feedback bandwidth but bringing drawbacks of necessary wavelengths control and extra detectors in Charlie's setup. With removal of the need for service fibre, we believe the benefit outweighs the complexity of extra detectors in Charlie's setup.

About expression of some sentences.

1) First paragraph in the second page (line 33), ‘but extracts information from single photon interference rather than two-photon coincidence’, most information is extracted from single photon, but we still can extract information from multi-photon states. Please give a more strict expression.

We thank the reviewer for pointing out the shortcoming in our description. We

intended to use the term “single photon interference” to refer the interfering signals are attenuated to a single-photon level, but it be confused as “single photons”. To be precise, we have revised the sentence, reading as “...but extracts the information from **the first-order interference** rather than two-photon coincidence.”

2) Equation (1)’s light speed in the fibre, and ΔL the length difference between the users’ fibres to Charlie’, I think the refractive index of the fiber is also one key factor that introduces differential phase. One better expression might be $\Delta(nL)$, the authors can see equation (1) in ref. 31.

The original Eq. (1) implicitly assumes a constant of fibre refractive index. While this assumption does not affect the approach or result of our manuscript, we understand the reviewer’s point that the change in refractive index is also a factor in the differential phase variation. Hence, we have revised Eq. (1) and its associated description by using the differential optical path length $\Delta(nL)$ to replace the term $n\Delta L$.

3) The second paragraph of supplementary information, ‘ IM_1 is modulated for pulse carving, IM_2 by a waveform signal for preparing pulses of different intensities’, and ‘The encoder is able to achieve >40 dB extinction ratio between the signal (μ_Z) and vacuum (μ_0) states.’ I think there might be some misleading expression about such high extinction ratio with only one intensity modulator. Please carefully check it.

We thank the reviewer for raising this point. It is important to achieve a high extinction ratio between the signal and vacuum states, so we use the first IM also to help improve this extinction ratio, *i.e.*, extinguishing the intensity of the vacuum state pulses during pulse carving. That is to say, we use first two IMs to achieve a more than 40 dB extinction ratio between the signal (μ_Z) and vacuum (μ_0) states.

Accordingly, we have revised the sentence to “ IM_1 is modulated for **carving out signal and decoy pulses and for extinguishing light transmission at vacuum time slots**, IM_2 by a waveform signal **for setting the signal/decoy intensity and further extinguishing the vacuum signals**, ...”.

4) In 5th page of supplementary information, the line numbers (99~112) overlap with TABLE S2, please correct the box of TABLE S2 in the revised version.

We have corrected Table S2.

REVIEWERS' COMMENTS

Reviewer #1 (Remarks to the Author):

The authors have successfully addressed all the comments I made in the previous round of review. The paper can be published in its current form.

Reviewer #2 (Remarks to the Author):

The authors satisfactorily and extensively replied to all of my concerns and also to those of the other reviewer.

The additional details in the revised manuscript and supplementary materials show more clearly that the work contains enough novelty, and more friendly to readers interested in quantum information. The impact of the work was already present in the previous manuscript and I already assessed it in my first report.

Therefore, I recommend the publication in its current form.